# Potential links between human bloodstream infection by *Salmonella enterica* serovar Typhimurium and international transmission to Colombia

Yan Li[1], Caisey V. Pulford[1], Paula Díaz[2], Blanca M. Perez-Sepulveda[1], Carolina Duarte[2], Alexander V. Predeus[1], Magdalena Wiesner[2], Darren Heavens[3], Ross Low[3], Christian Schudoma[3], Angeline Montaño[2], Neil Hall[3,4], Jaime Moreno[2], Jay C. D. Hinton[1]*

**1** Institute of Infection, Veterinary & Ecological Sciences, University of Liverpool, Liverpool, United Kingdom, **2** Grupo de Microbiología, Instituto Nacional de Salud, Bogotá, Colombia, **3** Earlham Institute, Norwich, United Kingdom, **4** School of Biological Sciences, University of East Anglia, Norwich, United Kingdom

* jay.hinton@liverpool.ac.uk

## Abstract

*Salmonella enterica* serovar Typhimurium is a prevalent food-borne pathogen that is usually associated with gastroenteritis infection. *S.* Typhimurium is also a major cause of bloodstream infections in sub-Saharan Africa, and is responsible for invasive non-typhoidal *Salmonella* (iNTS) disease. The pathogen also causes bloodstream infection in Colombia, but there has been a lack of information about the *S.* Typhimurium isolates that were responsible. Here, we investigated the genomic characteristics of 270 *S.* Typhimurium isolates from bloodstream infection patients in Colombia, collected between 1997 and 2017. We used whole-genome sequencing to analyse multidrug-resistant (MDR) profiles, plasmid distribution, and to define phylogenetic relationships. The study identified the distinct sequence types and phylogenetic clusters of *S.* Typhimurium prevalent in Colombia. The majority of isolates (90.8%) were ST19, which is distinct from the iNTS-associated *S.* Typhimurium in sub-Saharan Africa (ST313). The two prominent clusters of MDR *S.* Typhimurium were either DT104 or closely related to the LT2 reference strain. We used a phylogenetic approach to associate the Colombian clusters with global *S.* Typhimurium isolates from public databases. By putting the Colombian *S.* Typhimurium isolates in the context of the global spread of DT104, ST313 and LT2-related variants, we found that the Colombian clusters were introduced to the country via multiple independent events that were consistent with international transmission. We suggest that the acquisition of quinolone and chloramphenicol resistance by the Colombian *S.* Typhimurium isolates was driven by horizontal gene transfer. Three ST313 isolates that caused bloodstream infection in Colombia were identified. These ST313 isolates were related to the Malawian ST313 lineage 3 & UK ST313, and shared a similarly high invasiveness index. To our knowledge, this is the first report of ST313 in Colombia.

**Data availability statement:** Fastq files for the 270 *Salmonella* genomes are available at NCBI/EMBL under the accession numbers listed in S1 Table (Project PRJEB35182). The complete genome sequences of the novel plasmids identified in this study are available at NCBI (Project PRJNA1095721).

**Funding:** This work was supported by a Wellcome Trust Investigator award to JCDH (Grant number 222528/Z/21/Z), and by a Global Challenges Research Fund (GCRF) award to NH and JCDH (BBS/OS/GC/000009D). NH is supported by the BBSRC Core Strategic Programme Grant BB/CSP17270/1. DH is supported by the Earlham National Capability in Genomics (BB/CCG1720/1). The funders had no role in study design, data collection and analysis, decision to publish, or preparation of the manuscript.

**Competing interests:** The authors have declared that no competing interests exist.

## Author summary

*Salmonella* Typhimurium is a serovar of the bacterial species *Salmonella enterica*. *S.* Typhimurium usually infects humans through contaminated food or drink to cause self-limited diarrhoea. However, *S.* Typhimurium can also cause invasive non-typhoidal *Salmonella* (iNTS) disease, a dangerous bloodstream infection. In sub-Saharan Africa, most of the iNTS-associated *S.* Typhimurium isolates belonged to sequence type ST313. It was not known what type of *S.* Typhimurium caused iNTS in Colombia. Whole genome sequencing (WGS) has been widely used to monitor *Salmonella* infections in European and Northern American countries. However, WGS data from Latin American countries such as Colombia has been very limited. Here, we analysed 270 Colombian *S.* Typhimurium isolates from bloodstream infection patients with iNTS disease. Using WGS and bioinformatic analyses, we answered many questions about the Colombian *S.* Typhimurium isolates. The majority of the iNTS-associated isolates in Colombia belonged to ST19, rather than the ST313 sequence type responsible for bloodstream infections in sub-Saharan Africa. We compared the three ST313 isolates from Colombia and Malawi. The Colombian ST313 cluster was closely related to Malawian ST313 L3, and both appeared to have adaptations to an extraintestinal lifestyle. We suggest that the limited number of ST313 bloodstream infections in Colombia reflects the highly immunocompetent population, in contrast to the substantial numbers of immunocompromised people in sub-Saharan Africa. We identified ST19 multi-drug resistance clusters linked to globally-distributed *S.* Typhimurium isolates. One MDR cluster (Cluster 2) is likely to have spread from Europe or North America. We propose that a second MDR cluster (Sub-cluster 7.3) recently emerged in Colombia, and acquired plasmid-associated resistance genes prior to transmission to Ecuador.

## Introduction

*Salmonella enterica* serovar Typhimurium (*Salmonella* Typhimurium) is a major cause of non-typhoidal salmonellosis worldwide. In Colombia, laboratory-based surveillance by the Instituto Nacional de Salud involved 12,055 *S. enterica* isolates from 1997 to 2017, and found that *S.* Typhimurium was responsible for 28.4% of the cases identified by passive laboratory surveillance in the country. The majority (54.9%) of the 1,302 *S.* Typhimurium isolates were multidrug-resistant (MDR, defined by resistance to three or more antimicrobials), and 32.4% were resistant to one or two antimicrobial agents [1].

Over the past decade, whole-genome sequencing (WGS) has revolutionised our understanding of the epidemiology of *Salmonella*. Since 2014, WGS-based surveillance has been a routine procedure in the UK [2,3]. The availability of thousands of whole-genome sequences of *S.* Typhimurium has given epidemiological investigators an unprecedented ability to discover outbreaks, such as the international outbreak of *S.* Typhimurium ST34 [4,5]. Recent efforts have been made to systematically collect *Salmonella* Typhimurium genomic data from Africa and Asia [6,7]. However, compared to Europe and the US, there has been insufficient *Salmonella* genome data from low- and middle-income countries, including Latin American countries. In Colombia, the existing laboratory-based surveillance has yielded some insights, but critical questions remain unanswered. Specifically, there is an urgent need to explore the evolutionary lineages of multidrug-resistant (MDR) *S.* Typhimurium isolates, as well as the genetic determinants of antimicrobial resistance, and the horizontal gene transfer (HGT)

elements involved in the spread of resistance genes. An understanding of any international transmission that has influenced the spread of MDR isolates within and beyond Colombia could contribute to public health policy and interventions in the country.

Although usually associated with self-limiting diarrhoea, non-typhoidal *Salmonella* serovars can spread from the human intestine to normally sterile sites and develop bacteraemia, meningitis, and other focal infections, collectively referred to as Invasive Non-Typhoidal *Salmonella* (iNTS) disease [8]. In 2017, fatal iNTS bloodstream infections were responsible for more than 77,000 deaths worldwide [9]. In sub-Saharan Africa (SSA), iNTS disease has been associated with two distinct lineages of *S.* Typhimurium ST313, both carrying MDR-encoding Tn*21* elements on plasmid pSLT [10] and the prophages BTP1 and BTP5 [11]. A third pan-susceptible lineage of *S.* Typhimurium ST313 was found in Malawi recently, with a higher invasiveness index [12]. ST313 Lineage 3 was closely related to ST313 isolates responsible for gastroenteritis in the UK and Brazil [13,14]. However, while *S.* Typhimurium ST313 is known to be a major cause of iNTS disease in Africa [10,12], the role of ST313 in Colombia has been unclear.

The multidrug-resistant *S.* Typhimurium phage type DT104 emerged in the 1980s, and subsequently became the dominant *Salmonella* clone associated with infection in both agricultural animals and humans in European and North American countries [15]. DT104 isolates were resistant to ampicillin, chloramphenicol, streptomycin, sulphonamides and tetracyclines (ACSSuT resistance type) [16]. A WGS-based analysis revealed that DT104 initially acquired MDR determinants and was subsequently transmitted to American countries such as the US, Canada, and Argentina via separate routes [17]. Investigation of DT104 isolates sampled from human bloodstream and stool demonstrated that DT104 was no more invasive than other common *Salmonella* serovars [18].

The "10,000 *Salmonella* genomes project" was launched by a collaborative UK team as a global effort to generate *Salmonella* genomic information from 53 countries/territories, mostly low- and middle-income countries [19]. About 25% of the *Salmonella* isolates sequenced in the project originated from Latin American countries, including 270 *S.* Typhimurium and monophasic *S.* Typhimurium isolates from Colombia. Monophasic *S.* Typhimurium are variants of *S.* Typhimurium that do not express the flagellin phase 2 H-antigen [20]. The 270 isolates were isolated from the bloodstream of patients with systemic iNTS disease.

We used the genome sequences to perform phylogenetic analyses. This bioinformatic dissection of the 270 genomes involved the analysis of core genome SNPs, antimicrobial resistance (AMR) genes and plasmid profiles. We used maximum likelihood methods to investigate the evolutionary history of *S.* Typhimurium and monophasic *S.* Typhimurium in Colombia.

This study identified the sequence types, MDR profiles, and plasmid distribution of the *S.* Typhimurium isolates that cause bloodstream infection in Colombia and linked the Colombian clusters with public databases. We investigated the role of ST313 in iNTS disease in Colombia and compared the invasive potential of Colombian ST19 and ST313 isolates in comparison with African *S.* Typhimurium ST313. Public genome databases allowed us to put the Colombian *S.* Typhimurium and monophasic *S.* Typhimurium isolates into a global context, consistent with international transmission of Colombian ST313 and the two predominant MDR clades.

## Methods

### Ethics statement

This research has the approval of the Technical Research Committee (CTIN) and the Ethics Research Committee (CEIN) of the Instituto Nacional de Salud (INS), Colombia, with the codes CTIN-05–2015 and CTIN-05–2017.

## Bacterial isolation and characterisation

The *Salmonella* isolates investigated in this study were collected and characterised at the Instituto Nacional de Salud, Colombia. A collection of 270 *Salmonella* human blood isolates were used for this study, obtained between 1999 and 2017 from 22 of the 33 Colombian departments. Phenotypic serotyping was performed at the Instituto Nacional de Salud using the White-Kaufmann-Le Minor scheme [21].

AMR was determined phenotypically using the Kirby-Bauer test and the Minimum Inhibitory Concentration test by semi-automated MicroScan and Vitek 2 platforms, following the performance standards of the US Clinical and Laboratory Standards Institute [22]. The antimicrobials tested were ampicillin, chloramphenicol, streptomycin, tetracycline, gentamicin, amikacin, nalidixic acid, trimethoprim, ciprofloxacin, ceftazidime, and cefotaxime. The metadata of all isolates is summarised in S1 Table, including serovars, collection date, location, isolation source, and phenotypic AMR profile.

## Whole genome sequencing and assembly

DNA extraction and whole genome sequencing were carried out at the Earlham Institute, Norwich, UK, as part of the 10,000 *Salmonella* Genomes consortium [19]. The adapter sequences of Illumina raw reads were trimmed using Trimmomatic v0.36 [23] in palindrome mode (ILLUMINACLIP:2:30:10). Quality trimming was conducted by Seqtk v1.2-r94 [24] using Phred algorithm. The genome assembly was performed with Unicycler v0.4.4 [25] with paired reads. The quality of assemblies was assessed by reference-based Quast v4.6.3 [26]. A prophage and plasmid-free version of the *Salmonella* Typhimurium ST4/74 genome (GenBank ID: CP002487) was used as the Quast reference, in order to exclude the impact of variable regions of the genome on quality assessment. N50 values and the number of contigs were evaluated. The N50 values of all assemblies were >20 kb, and the number of contigs was <500.

## Genomic prediction of MLST, serovar, AMR and plasmid profile

MLST v2.10 [27] was used for *in silico* sequence typing from raw reads and assembled data. MLST used the PubMLST database [28] and based on the multi-locus sequence typing (MLST) scheme of *S. enterica*, which defined the *Salmonella* sequence type on the basis of 7 housekeeping genes [29]. SISTR v1.1.1 [30] was used for the *in silico* serovar prediction. The monophasic variants of *S.* Typhimurium were then defined by the absence of *fljAB* genes. Ariba v2.14.6 [31] was used to identify the presence or absence of *fljAB* gene from raw reads. AMR genes within the assemblies were identified using Abricate v1.0.1 [32] with the Resfinder (https://cge.cbs.dtu.dk/services/ResFinder/) [33] database (coverage >70%). The accuracy of genotype-based predictions of antimicrobial resistance was assessed using sensitivity and specificity, and was calculated from a confusion matrix. Sensitivity was defined as True Positive/ (True Positive + False Negative), and specificity as True Negative/ (True Negative + False Positive).

To infer the presence or absence of the *S.* Typhimurium virulence plasmid pSLT, the reads of all the isolates were mapped to the reference pSLT sequence from *S.* Typhimurium LT2 (GenBank ID: NC_003277.2) using BWA-MEM v0.7.17 [34]. The read depth of each site of pSLT was then summarised by Samtools v1.15 [35]. The regions of the plasmid with more than 5x depth were marked as aligned in the isolate, otherwise thought to be missing. The aligned length of pSLT were then calculated in percentage.

To investigate the plasmids other than pSLT within the Colombian *S.* Typhimurium population, the circular contigs from all assemblies were extracted using SeqKit v2.0.0 [36]. Similar circular contigs were then clustered using Mash v2.3 with thresholds of maximum distance

0.05 and maximum p-value 0.05 [37]. The representative contigs from each cluster were then compared to a plasmid database PLSDB (https://ccb-microbe.cs.uni-saarland.de/plsdb/) [38] using BLASTn v2.12.0 [38,39]. The BLASTn hits were filtered with at least 90% identities. If the overall BLAST coverage of a contig was 100% against the reference plasmid, it was marked as "identical". Otherwise, if a contig was partially aligned, it was marked as "similar". Only the circular contigs that were identical or similar to a known plasmid were identified as plasmids. The pMLST type of each plasmid cluster was identified using Abricate v1.0.1 [32] with Plasmid-Finder [40] database. If the plasmid was identical to a reference in PLSDB, the name of the reference was used. Otherwise, the plasmid was named in the form of "pCOT1_IncX1_158-JM", where "pCOT1" represents "plasmid Colombian Typhimurium 1", "IncX1" indicates its pMLST type if available, and "158-JM" refers to the strain harbouring the plasmid.

## Phylogenetic analysis

A core genome alignment was built from the Colombian *S.* Typhimurium isolates. Annotation of the genome assemblies was done with Prokka v1.13 [41]. The pan-genome analysis program Panaroo v1.2.9 [42] was then used to conclude the core genes and build the core genome alignment from the genome annotation. In Panaroo, core genes were defined as genes that existed in more than 95% of the genomes.

To contextualize the Colombian *S.* Typhimurium isolates, 22 pre-existing genomes (S2 Table) were downloaded from RefSeq and included in the core genome alignment. Panaroo generated the core genome alignment from 4170 core genes. A maximum-likelihood phylogenetic tree was inferred with RAxML-NG v0.6.0 [43] using the GTR+G model from the core genome alignment with 100 bootstrap iterations. The tree was rooted using *S.* Typhi strain CT18 as an outgroup (GenBank accession number: AL513382.1). The core genome maximum likelihood tree was visualised with iTOL [44].

The HierCC (Hierarchical Clustering of cgMLST) clusters of the genomes were assigned with Enterobase [45]. HierCC is a system that classifies bacterial genomes into different levels of clusters based on the cgMLST allele distances. For example, the genomes within the same HC100 cluster have fewer than 100 allele differences [45]. The Colombian *S.* Typhimurium clusters were defined at the HC100 level. For Cluster 7, three sub-clusters were identified at the HC50 level. The transmission history of three Colombian clusters was traced in this study, including cluster 2, cluster 3 and sub-cluster 7.3, which belong to HierCC clusters HC100_20801, HC100_305 and HC50_64733, respectively. The *Salmonella* genomes within the same clusters from the other countries were queried through Enterobase [45,46]. For the HC100_20801 and HC50_64733 clusters, all the genomes in the same cluster were downloaded from Enterobase. The HC100_305 cluster associated with the DT104 global epidemic from the 1990s contained 6,616 isolates, 77.6% of which were from Europe (n = 3405) and the US (n = 1729) (http://enterobase.warwick.ac.uk/, accessed 04 Jan 2022). To avoid sampling bias and emphasize the Latin American isolates, all the genomes of Latin American HC100_305 cluster isolates were downloaded from Enterobase, while genomes from other continents were downloaded from a former study which summarised the DT104 global transmission [17].

The core genome maximum-likelihood phylogenetic tree of each cluster was constructed using Prokka, Panaroo and RAxML-NG as described above. The core genome alignment of HC100_305, HC100_20801, HC50_64733 clusters were generated by Panaroo from 4463, 4456, 4311 core genes respectively. The genome of the *S.* Typhimurium isolate ST4/74 (GenBank accession number: GCF_000188745.1) was used as an outgroup. The Bayesian inference of ancestral dates was performed by BactDating v1.1.0 [47] from the phylogenetic tree.

The geographical transmission history was inferred by the make.simmap function in the R package phytools [48,49]. The make.simmap function simulates the stochastic character map while treating the continent of origin as a discrete character trait of the tree tips. The geographical transmission history was summarised from 100 simulations using the Empirical Bayes method of the make.simmap function. Thus, the posterior probabilities of the ancestral nodes were estimated and visualised. The location of the ancestral nodes was determined with posterior probability >50%. To estimate the year range of the transmission event, the beginning year is the median value of the node before being introduced to Colombia, and the final year is the median value of the MRCA of the Colombian isolate. The transmission history was then visualised on the world map using Plotly in Python.

## Invasiveness index

The invasiveness index is an assessment of the adaptation of *Salmonella* to an invasive lifestyle based on a machine learning model [50]. The training set of the model was a set of reference genomes from gastrointestinal and extraintestinal *Salmonella* serovars. Delta-bitscore was used to measure the functional consequence of the gene mutations in each genome [51]. The inputs were trained by the random-forest algorithm. As a result, 196 predictor genes showed a strong correlation with invasiveness, most of which were degraded or absent metabolic genes in extraintestinal strains. The machine learning model was validated with the *S.* Enteritidis and *S.* Typhimurium genomes that cause iNTS disease in sub-Saharan Africa [12,50]. Here, the invasiveness index of each genome assembly was calculated from the pre-trained model [50]. A box plot was made using the R package ggplot2 to compare the invasiveness index of each cluster [52].

# Results

## Phylogenetic analysis and metadata

To determine the genetic structure of *S.* Typhimurium and the monophasic *S.* Typhimurium variants responsible for iNTS disease in Colombia, we sequenced the genomes of 270 *S.* Typhimurium and monophasic *S.* Typhimurium isolates collected from the blood of patients. Twenty-two publicly available genomes from the RefSeq database were used as contextual references, including the well-characterised *S.* Typhimurium genomes LT2, ST4/74, 14028s, D23580 etc (Accession IDs in S2 Table). Core genome analysis identified 3,777 core genes. A total of 53,228 SNPs were identified from the core gene alignment. A phylogenetic tree was constructed with the maximum likelihood method (Fig 1). The phylogenetic tree demonstrated robust support for the major clades (bootstrap > 80%) (S1 Fig).

Monophasic *S.* Typhimurium are variants of *S.* Typhimurium that do not express the phase 2 H-antigen [20]. The absence of phase 2 H-antigen in monophasic *S.* Typhimurium is caused by deletion of the *fljAB* operon [20]. To distinguish the monophasic *S.* Typhimurium variants from the other *S.* Typhimurium isolates, we compared three different methods: phenotypic determination using White-Kauffmann scheme, *in silico* prediction by SISTR, and the presence or absence of the *fljAB* genes. Phenotypic serotyping identified 22 monophasic *S.* Typhimurium isolates, which showed inconsistency in phylogenetically closely-related isolates. SISTR failed to identify all the isolates in the monophasic lineage 4, which were discovered by the other two methods. Based on the absence of *fljAB* genes, 45 monophasic isolates were identified and aligned well with the phylogenetic analysis. The four monophasic *S.* Typhimurium lineages were marked by numbers in circles in Fig 1. The results of the three identification methods are detailed in S1 Table and S1 Fig.

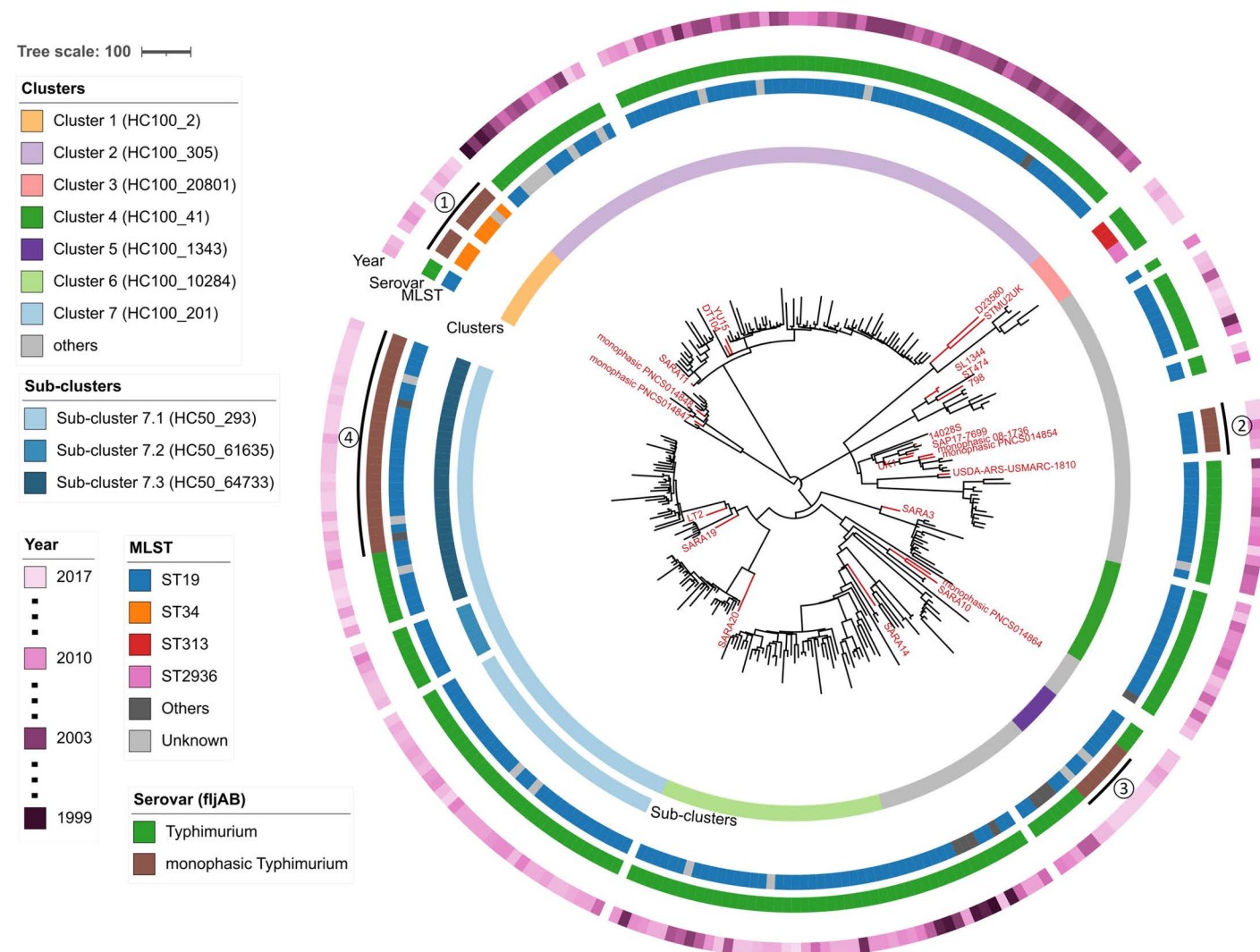

**Fig 1. Maximum-likelihood phylogeny of 270 *S.* Typhimurium and monophasic *S.* Typhimurium isolates from Colombia.** Maximum likelihood phylogeny demonstrating the population structure of *S.* Typhimurium ST19, ST34, ST313 and other sequence types. Twenty-three contextual *S.* Typhimurium genomes are shown as red branches with red labels for each. The coloured rings indicate, from inside to outside, the clusters (defined at the HC100 level), sub-clusters (defined at the HC50 level), serovars (defined by the presence or absence of *fljAB* genes), sequence types, and year of sampling. The monophasic *S.* Typhimurium lineages are numbered 1, 2, 3 and 4. The scale bar represents the number of SNPs per branch. The isolate names and the position of outgroup *Salmonella* Typhi strain CT18 are shown in the S1 Fig.

Although SISTR is commonly used for *in silico* serovar prediction, the software tool could not identify certain lineages of monophasic *S.* Typhimurium. SISTR predicts serovar from the combination of the antigen gene and cgMLST analysis, using 330 loci [30]. However, antigen gene BLAST results can yield multiple possible serovars due to incomplete assembly of contigs found in Illumina short-read sequence data. The cgMLST method relies on a database of 330 core genes selected from over 4,000 publicly accessible *Salmonella* genomes [30]. As a result, it is theoretically difficult to identify newly emerged monophasic *S.* Typhimurium lineages that lack representative genomes in the SISTR database, because new monophasic lineages may share similar core gene profiles with the ancestral *S.* Typhimurium.

The sequence types of the *S.* Typhimurium and monophasic *S.* Typhimurium isolates were determined by MLST (S1 Table). The STs of 20 isolates were unknown because at least one of the seven housekeeping genes used for the MLST analysis was incomplete following genome assembly. The majority (90.8%, 227/250) of the isolates were ST19. We found three ST313 isolates closely related to two ST2936 isolates, while ST2936 is a single locus variant of ST313 that differs by a single MLST allele (*sucA*). To our knowledge, this is the first report of ST313 in Colombia, which is significant because ST313 is a major cause of iNTS disease in Africa.

An ST34 clade of eight isolates was associated with the monophasic *S.* Typhimurium Lineage 1. ST34 was first reported in Europe, and was also responsible for iNTS disease in Vietnam [53,54]. Coupled with the previous report of two Colombian *S.* Typhimurium ST34 isolates that harboured the *mcr-1* gene [55], our finding provides evidence that ST34 is responsible for a subset of human bloodstream infections in Colombia.

The clusters of the isolates were defined using the Enterobase HierCC system [45] at the level of 100 alleles (HC100). Seven of the clusters were labelled, and the other smaller clusters were included as "others" (Fig 1). The two main clusters with the greatest number of isolates were Cluster 2 (HC100_305) and Cluster 7 (HC100_201). Cluster 2 was associated with phage type DT104. Most of the Cluster 2 isolates were isolated before 2010 (88%, 66 out of 75). The contextual reference DT104 was isolated in Scotland, associated with MDR and widespread zoonotic infection [56]. DT104 is a multi-resistant phage type responsible for global epidemics from the 1990s [17,56,57].

Cluster 7 was linked to the *S.* Typhimurium reference genome LT2. Most of the Cluster 7 isolates were collected after 2010 (94%, 66 of 70). Cluster 7 was further divided into three sub-clusters using HierCC at the level of 50 alleles (HC50). Sub-cluster 7.3 included closely related monophasic *S.* Typhimurium and wild-type *S.* Typhimurium isolates, consistent with the recent deletion of *fljAB* genes.

Cluster 1 (HC100_2) included the eight monophasic *S.* Typhimurium ST34 isolates and two *S.* Typhimurium ST19 isolates. Cluster 3 (HC100_20801) included three ST313 isolates and two ST2936 isolates. As ST313 was the major cause of the iNTS epidemic in sub-Saharan Africa [12,58], a genomic comparison of Colombian ST313 and African ST313 is presented below.

## AMR genotype and phenotype

*Salmonella* genomes are increasingly being used to predict AMR characteristics. For example, a massive study validated 97.8% of genome-based AMR predictions involving 15 antimicrobials in 3,491 non-typhoidal *Salmonella* isolates that belong to 227 serovars [59]. We predicted the AMR profiles of the Colombian *Salmonella* isolates from WGS data and compared the results to the phenotypic antimicrobial susceptibility tests involving a panel of 11 drugs. In summary, the most common resistance phenotype involved streptomycin, with 61.9% of the isolates being resistant or intermediately resistant to streptomycin (n = 167), followed by tetracycline resistance (41.9%, n = 155), ampicillin resistance (32.6%, n = 88) and chloramphenicol resistance (31.5%, n = 85, S1 Table). The observed susceptibility phenotypes demonstrated strong concordance with the genotypic profiles. For instance, resistance to ampicillin was associated with the presence of resistance genes such as $bla_{CARB-2}$, $bla_{CMY-2}$, $bla_{CTX-M-12}$, $bla_{CTX-M-15}$, $bla_{OXA-1}$, $bla_{OXA-2}$, $bla_{OXA-392}$, and $bla_{TEM-1B}$. Predictions of ampicillin resistance based on the presence of these genes exhibited a sensitivity of 89.2% and a specificity of 95.2% (S1 Table).

Fig 2 compares the antimicrobial-resistant phenotypes and genotypes grouped into six categories: aminoglycoside, β-lactam, quinolone, chloramphenicol, tetracycline, and trimethoprim. The AMR genes associated with drugs not tested in this study were classified as

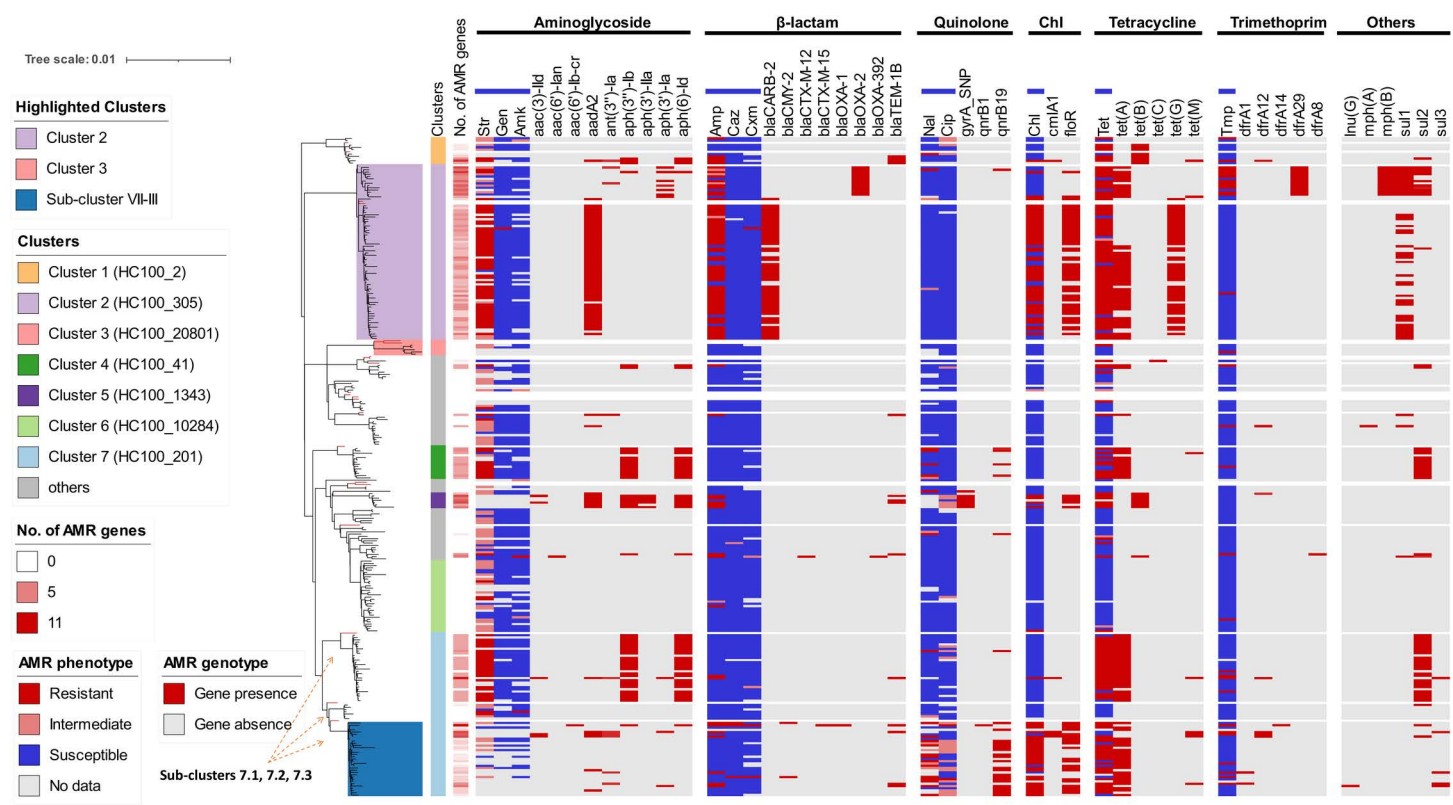

**Fig 2. Linking genotypes with the antimicrobial resistance phenotypes of Colombian *S.* Typhimurium isolates.** The maximum-likelihood tree is a linear version of the circular tree shown in Fig 1. On the phylogenetic tree, three highlighted clusters are shaded by coloured backgrounds and the eight cluster designations are indicated with coloured strips. In the vertical columns to the right, the number of AMR genes from 0 to 11 are shown in different shades of red. The AMR profiles are visualised as heatmaps. The results of phenotypic antibiotic susceptibility tests are highlighted with a **blue bar** above the relevant columns. The red, pink, blue and grey results of the antibiotic susceptibility tests are detailed in the key. In other columns, the genome-predicted presence and absence of AMR genes are indicated in red and grey, respectively. Antibiotics abbreviation: Amp, ampicillin; Chl, chloramphenicol; Str, streptomycin; Tet, tetracycline; Gen, gentamicin; Amk, amikacin; Nal, nalidixic acid; Tmp, trimethoprim; Cip, ciprofloxacin; Caz, ceftazidime; Cxm, cefotaxime. The 11 antibiotics are classified into six groups by the ResFinder database according to the drug family [33]. There are six AMR genes associated with antimicrobials that were not phenotypically tested in this study.

"Others", which included the *mph(B)* gene that encodes resistance to azithromycin and the *sul123* genes associated with sulphonamide resistance. Azithromycin and sulphonamide susceptibility were not phenotypically assessed in this study.

Isolates defined as MDR showed phenotypic resistance to at least three drugs tested in this study. A total of 114 out of the 270 *S.* Typhimurium isolates (42.2%) were MDR. Most of the MDR isolates belonged to the Clusters 2, 4, 5 and 7. The MDR Cluster 2 was divided into two patterns of AMR gene conservation: A sub-cluster with resistance to ampicillin, chloramphenicol, streptomycin, and tetracycline was predicted to be resistant to sulphonamide (ACSSuT type). This resistance type was similar to the MDR DT104 that emerged globally in the 1990s, which also carried the *aadA, floR, sul1* and *tet(G)* genes [17]. The other sub-cluster was resistant to ampicillin, tetracycline, trimethoprim, azithromycin and sulphonamide.

Cluster 7 was divided into three sub-clusters with distinct AMR profiles. Sub-cluster 7.1 was resistant to streptomycin, tetracycline and sulphonamide. Sub-cluster 7.3 was highlighted for carrying the quinolone-resistant gene *qnrB19*. Antimicrobial susceptibility testing confirmed that *qnrB19* encodes the resistance to nalidixic acid and is intermediately resistant to ciprofloxacin. Fluoroquinolone-resistant *Salmonella* has been listed as a high priority by the

latest WHO Bacterial Priority Pathogens List [60]. The *qnrB19*-harbouring Sub-cluster 7.3 also carried chloramphenicol resistance, encoded by *floR*, and tetracycline resistance encoded by *tet(A)*.

The ST313-associated Cluster 3 in Colombia was pan-susceptible. Because the ST313 isolates in Africa were multidrug-resistant [58] and the Colombian ST313 isolates shared phenotypic similarities in antibiotic-susceptibility with the UK variant and Lineage 3 of ST313 [14], we did the comparative genomic analysis detailed below.

Cluster 2 and Sub-cluster 7.3 had distinctive AMR profiles, representing the majority of MDR *S.* Typhimurium isolates in Colombia, highlighted in Fig 2. The phylogeny and evolution of these two (Sub-)clusters are shown below.

Although most of the antimicrobial resistance phenotypes were linked to gene presence, the reduced susceptibility to ciprofloxacin of Cluster 5 isolates was associated with point mutations in the *gyrA* gene (GyrA p.S83F or p.D87Y). Cluster 5 was also resistant to ampicillin, chloramphenicol and tetracycline. Genomic evidence suggests that Cluster 5 is resistant to nalidixic acid (*gyrA* mutation) and streptomycin (*aac(3), aadA2* and *aph*), however, this remains to be established phenotypically.

As well as the clade-specific antimicrobial resistance mentioned above, sporadic isolates exhibited resistance to third-generation cephalosporins (3GCs), including ceftazidime and cefotaxime. This resistance was likely conferred by the Extended-Spectrum Beta-Lactamase (ESBL) genes, such as $bla_{CMY-2}$, $bla_{CTX-M-12}$, and $bla_{CTX-M-15}$. The emergence of 3GC-resistant isolates is noteworthy, particularly because 3GC-resistant Enterobacterales are classified as a critical priority on the WHO Bacterial Priority Pathogens List [60]. Fortunately, carbapenem resistance, a critical priority of the WHO Bacterial Priority Pathogens List, was not detected in the Colombian *S.* Typhimurium dataset.

## Plasmid profile of Colombian *S.* Typhimurium

Important *Salmonella* phenotypes are encoded by plasmids, including resistance to antibiotics and heavy metals, utilisation of carbon sources, and virulence factors [61]. Previous studies showed that ~90% *S.* Typhimurium isolates carried a ~90 kb virulence plasmid, represented by the 94 kb pSLT in *S.* Typhimurium LT2 [62–65]. The pSLT plasmid is confined to the *Salmonella* serovar Typhimurium, is transferred vertically rather than horizontally [65,66], and encodes the *spv* virulence operon which promotes proliferation within mammalian cells [64]. In the Colombian *S.* Typhimurium isolates, the aligned percentage of the pSLT plasmid in each isolate was identified using the read mapping method. The results show that most of the Colombian *S.* Typhimurium and monophasic *S.* Typhimurium isolates carried a pSLT or pSLT-like plasmid, except Cluster 1 and Cluster 4 which lacked pSLT (Fig 3).

Non-pSLT plasmids were extracted from single circular contigs assembled from Illumina reads. By comparison to a manually corrected database, we identified plasmids with more than 80% BLAST coverage against plasmids from other hosts, including other genera such as *E. coli* and *Klebsiella pneumoniae*, or other *Salmonella* serovars. The high similarity of plasmids suggested that the small plasmids had been horizontally-transferred. GenBank entries of the reference plasmids are summarised in S2 Table.

Two small plasmids that carried AMR genes were found in Sub-cluster 7.3. The quinolone resistance gene *qnrB19* was encoded by a ~2,700 bp ColRNAI plasmid, which was 97.6% identical at the nucleotide level to pMK100, a 2,699 bp plasmid that had previously been identified in a Colombian *Salmonella* Infantis strain SO20, sampled from retail chicken in 2004 (GenBank ID: HM070379.1) [67]. The chloramphenicol resistance gene *floR* was carried by a novel 4,284 bp plasmid denoted as "pCOT7_GMR-S-276-JM". Further inspection showed

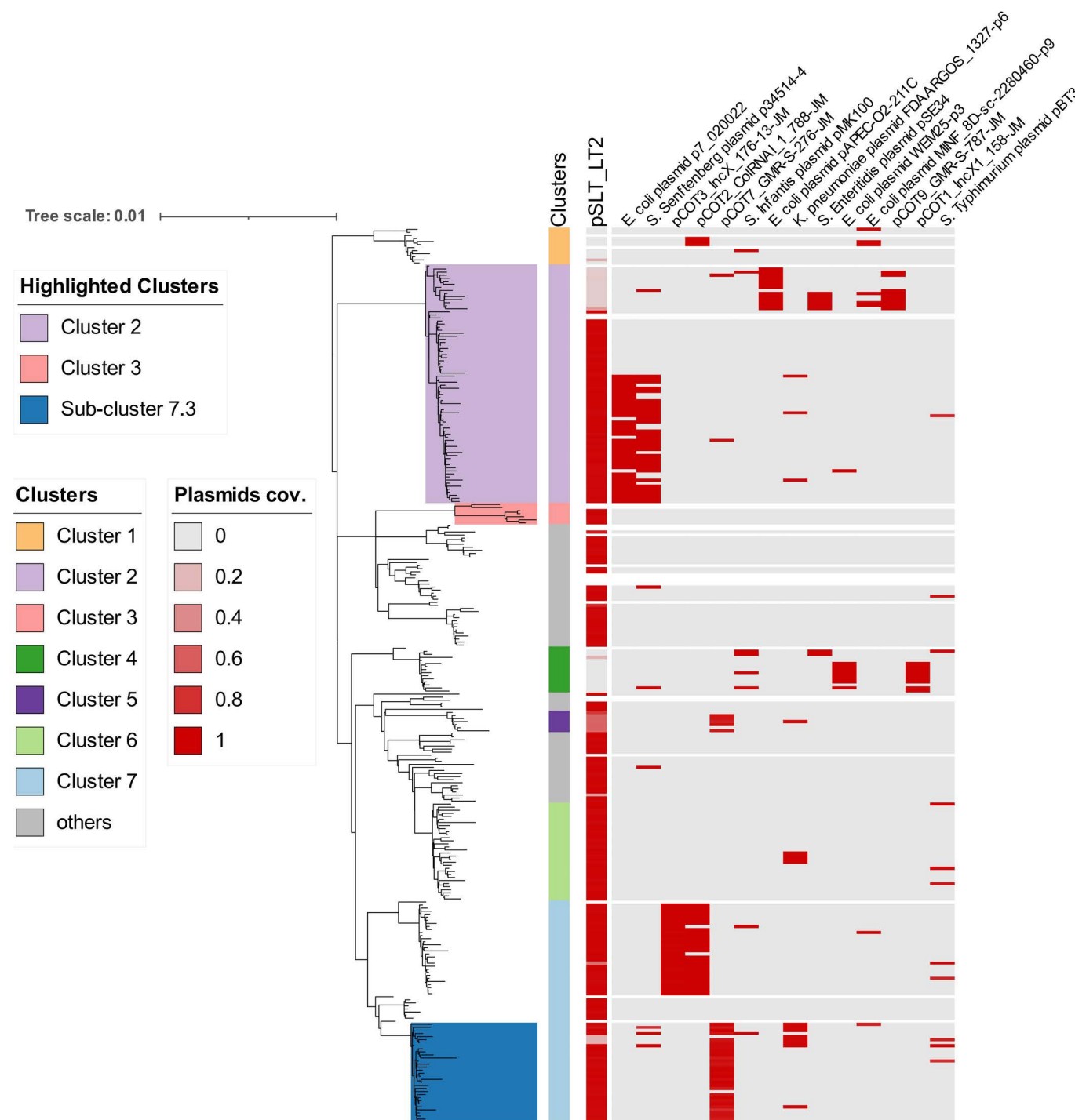

**Fig 3. The prevalence of the pSLT virulence plasmid and the distribution of small plasmids in the Colombian *S.* Typhimurium isolates.** The maximum-likelihood tree is the linear version of the circular tree shown in Fig 1. Clusters 2, 3 and sub-cluster 7.3 are highlighted on the tree. All clusters are shown as a vertical colour strip, as described in the key. The sequence coverage of the *S.* Typhimurium LT2 pSLT plasmid and 14 other small plasmids in the *S.* Typhimurium isolates is shown in gradations of Red. Only small plasmids with sequence coverage of >80% are shown in the figure. The names of the small plasmids were derived from either an identical reference plasmid from PLSDB or according to their plasmid-derived pMLST designation [40].

that the *floR* and *qnrB19* genes were integrated into linear contigs in multiple Colombian *S.* Typhimurium genomes, suggesting that the AMR genes had continued to evolve after being transferred into the genomes via small plasmids.

## The transmission history of three key clusters of *S.* Typhimurium

Three Colombian *S.* Typhimurium clusters were selected for further investigation of transmission history: Cluster 2 and Sub-cluster 7.3 were selected on the basis of AMR profiles and because they represented a significant part of the MDR isolates; Cluster 3 was selected to investigate association with the global ST313 isolates. Cluster 2 was the most prevalent in the dataset, associated with MDR DT104. Sub-cluster 7.3 carried quinolone and chloramphenicol resistance. Quinolone has been listed as the highest priority by the WHO in combatting antimicrobial resistance (World Health Organization, 2017). Cluster 2 and Sub-cluster 7.3 included 77% (88/114) of the MDR strains. Cluster 3 represented ST313 isolates, a sequence type associated with iNTS disease in sub-Saharan Africa.

To determine the geographical origin and infer the international transmission history of the *S.* Typhimurium clusters, *S.* Typhimurium genomes that belonged to the same HierCC clusters and originated from a variety of countries were downloaded from the public Enterobase genome resource (https://enterobase.warwick.ac.uk/). Bayesian analysis was used to estimate the dates of the nodes in the phylogenetic trees. The transmission routes for the *S.* Typhimurium clusters were inferred by stochastic mapping of the geographic origin of each isolate onto the tree [48].

Cluster 2 was associated with the phage type DT104 and the Enterobase HC100_305 cluster. DT104 caused a global epidemic in the 1990s and 2000s [17,56,57]. Due to the high number of HC100_305 genomes in Enterobase (>6,000 genomes, accessed 04 Jan 2022) and the significant sampling bias (77.6% from Europe and the US), a subset of public genomes from different countries was used to provide context for our study. Samples from Europe, North America, and Asia were downloaded from a previous study of the global transmission of DT104 [17]. The genomes of all Latin American HC100_305 isolates were downloaded from Enterobase. In total, 158 publicly-available DT104 genomes were selected (S3 Table).

The data suggested that Cluster 2 isolates were transmitted to Colombia through three independent introduction events: 1968–1975 from Asia, 1980–1984 from North America, and 1992–1994 from Europe (Figs 4A and 5A). The analysis indicated that the ancestors of the Asian and North American HC100_305 isolates likely originated from Europe over different time periods.

Cluster 3 (HC100_20801) is associated with *S.* Typhimurium ST313, the sequence type linked to iNTS disease in SSA [68]. ST313 is divided into three SSA lineages and several UK and Brazilian lineages [12–14,58,]. For our study, 87 contextual ST313 genomes were downloaded from the HC100_20801 cluster from Enterobase (S3 Table). We found that the HC100_20801 cluster included the Colombian ST313 isolates, the UK-ST313 [14], the Malawian ST313 Lineage 3 [12], and other ST313 isolates from Africa, North America and South America.

The geographic distribution of the isolates identified HC100_20801 as an ST313 sublineage that has been transmitted globally. The phylogenetic analysis revealed that the common ancestor of HC100_20801, the global ST313 sublineage, originated in Europe, specifically in the UK. Subsequently, HC100_20801 was disseminated to Africa, North America, and South America independently. The previously reported Malawian ST313 Lineage 3 was part of the African clade [12] and was estimated to be introduced to Malawi from South Africa between 1980–1993.

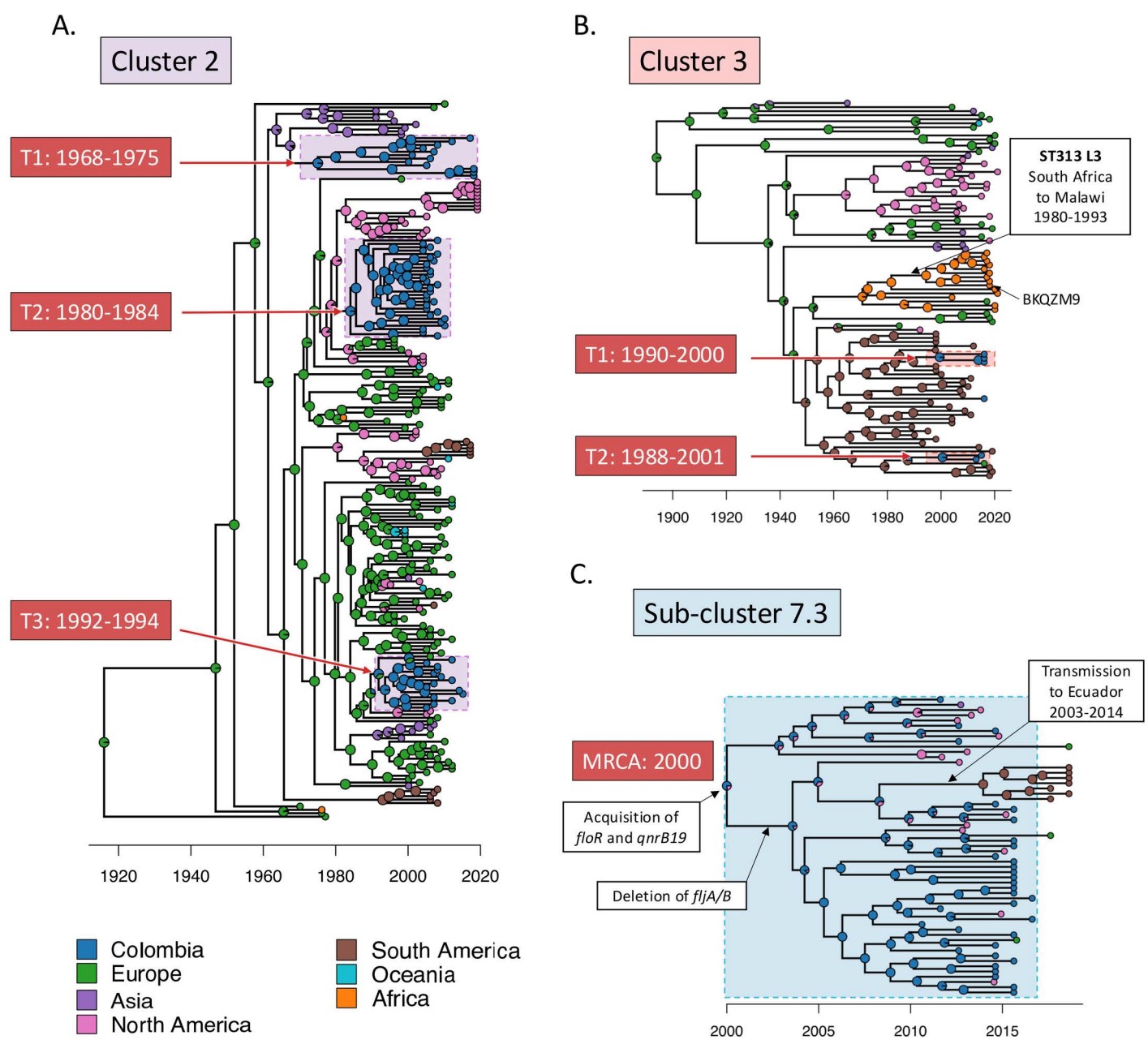

**Fig 4. Temporal and geographic transmission history of the global *S.* Typhimurium isolates belonging to three Colombian clades.** Global transmission events inferred from the Bayesian evolutionary analysis by BactDating v1.1.0 [47]. Three Colombian *S.* Typhimurium clades from Figs 2 and 3 are highlighted with coloured backgrounds. The colours of tips and nodes represent geographic locations. When a node has two colours, the percentage of the colours represents the inferred likelihood of the node being associated with two geographic locations. Geographic transmission routes were determined with Make.simmap, as described in the Methods. When the MRCA of a clade had >50% possibility of being located in Colombia, the clade was designated as being transmitted in Colombia shown by coloured backgrounds on the tree. The date range of the introduction events is estimated from the year of the MRCAs before and after transmission to Colombia. (A) Three introduction events (T1, T2 and T3) of Colombian HC100_305 isolates. (B) A global HC100_20801 (ST313) lineage and two introduction events (T1, T2) to Colombia. The analysis suggests that ST313 L3 was introduced to Malawi from South Africa. The representative isolate of ST313 L3 (BKQZM9) is labelled. (C) HC50_64733 originated from Colombia and was introduced to Ecuador and sporadically transmitted to North America and Europe. The important evolutionary events are labelled in white boxes.

## A. Cluster 2 (HC100_305)

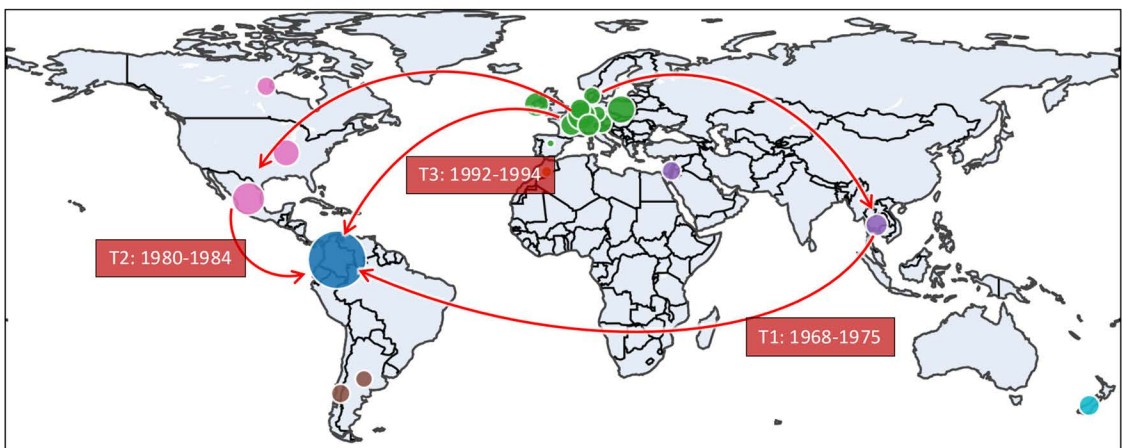

## B. Cluster 3 (HC100_20801)

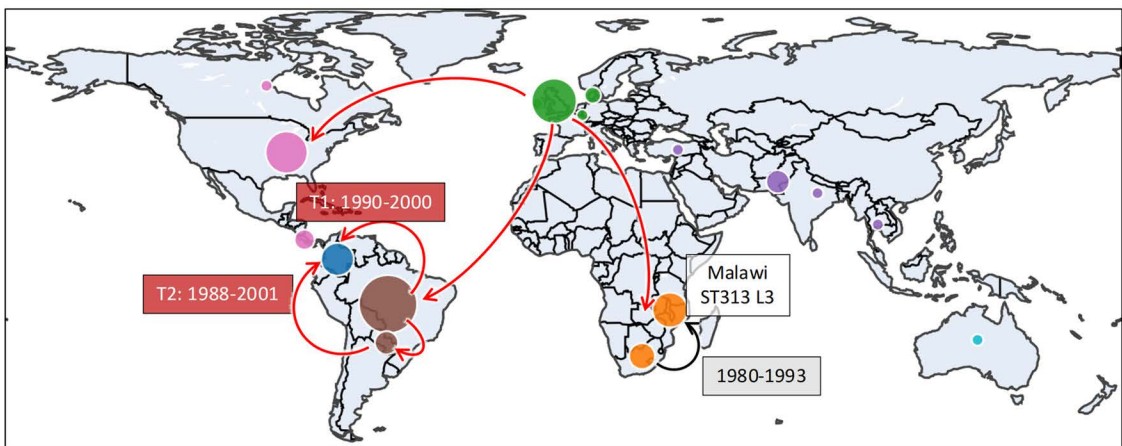

## C. Sub-cluster 7.3 (HC50_64733)

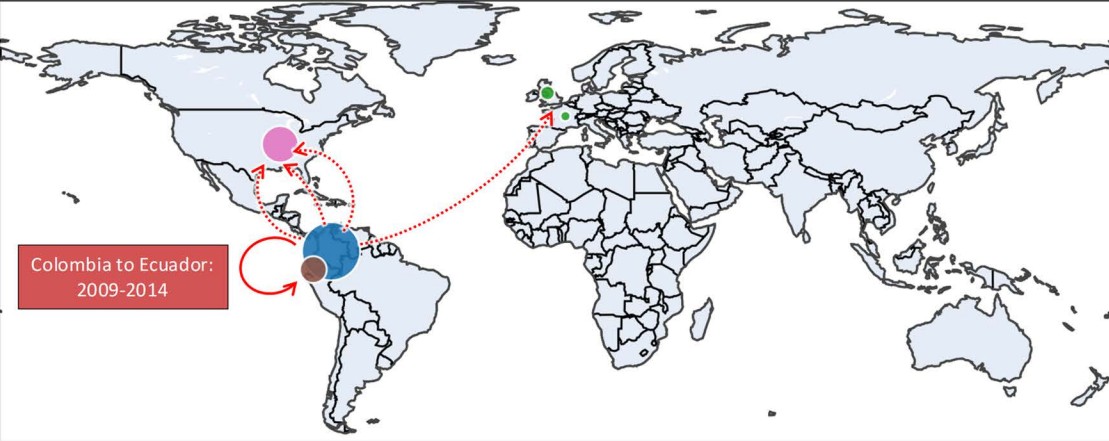

**Fig 5. Inferred international transmission routes of three Colombian clades.** The sample size of *S.* Typhimurium genomes from each country is shown as coloured bubbles. The lines show the likely transmission routes. The inferred years of introduction to Colombia are labelled with red text boxes. The dotted lines show sporadic transmissions. The base layer of the map was sourced from Natural Earth in GeoJSON format, available at https://geojson-maps.kyd.au/, and is provided under the public domain terms outlined at Natural Earth Terms of Use.

The Colombian ST313 Cluster 3 isolates, together with the Brazil ST313, formed the South American clade. Two introduction events to Colombia occurred concurrently: ST313 isolates arrived from Brazil between 1990–2000, while the ST2936 isolates (single-locus variants of ST313) arrived from Brazil or Paraguay between 1988–2001 (Figs 4B and 5B).

The Sub-cluster 7.3 carried a characteristic *qnrB19* gene that encodes resistance to quinolone. Previously, four *Salmonella enterica* isolates, including two *S.* Uganda, one *S.* Infantis and one *S.* 6,7:d:-, have been reported to carry a plasmid-mediated *qnrB19* gene in Colombia [67]. Sub-cluster 7.3 is the first report of *qnrB19*-harbouring *S.* Typhimurium isolates in this country. The cluster also carried the chloramphenicol-resistance gene *floR*.

We used the Enterobase database to put the Colombian Sub-cluster 7.3 into a global context by comparison with the HC50_64733 cluster. According to the phylogeographic analysis, the HC50_64733 cluster originated in Colombia and was subsequently transmitted to Ecuador between 2009–2014. Several additional independent sporadic transmission events to the United States and Europe were identified through the analysis (Figs 4C and 5C).

## The invasive potential of the *S.* Typhimurium lineages

This study focused on the genomes of *S.* Typhimurium isolates that cause bloodstream infection. However, the lack of animal models that faithfully recapitulate human invasive infection [69] has prevented meaningful experimental comparisons between the bloodstream and gastrointestinal isolates in the context of iNTS disease. Therefore, a machine learning-based prediction, Wheeler's invasiveness index [50], was used to compare the invasive potential of *S.* Typhimurium from each HC100 cluster.

The "invasiveness index" is based on the level of degradation of 196 predictor genes from invasive serovars [50]. A high invasiveness index is a signal that bacteria are adapting to an extraintestinal lifestyle. The iNTS-associated African *S.* Typhimurium ST313, which includes the lineages L1, L2 and L3, is in the process of adaptation to systemic infection and has higher invasiveness indices than ST19 [12,50]. The invasiveness index data in this study are detailed in S1 Table and summarised in Fig 6.

The important *S.* Typhimurium cluster in Colombia, Cluster 3 (ST313), had the highest invasiveness index (median = 0.191, s.d. = 0.011) (Fig 6). The value was similar to the invasiveness index of ST313 Lineage 3 in Malawi [12], consistent with the Malawian isolates belonging to the same HC100 cluster.

Cluster 5 showed the second highest invasiveness index (median = 0.158, s.d. = 0.015), with a value higher than that of the ST313 Lineage 1 (median = 0.129, s.d. = 0.008) and Lineage 2 (median = 0.134, s.d. = 0.006) [12]. Cluster 5 is the monophasic *S.* Typhimurium ST19 cluster with multiple drug resistances.

The invasiveness indices of the remaining five clusters, including the DT104-associated Cluster 2, were around 0.1, which is comparable to ST19 isolates from our previous study (median = 0.110, s.d. = 0.017) [12]. The result is consistent with the report that the proportion of DT104 isolates that cause either blood infection or gastroenteritis is similar to other non-invasive *Salmonella* serovars [18].

## Discussion

This study focused on genome data derived from *S.* Typhimurium isolates responsible for bloodstream infections in Colombia. Unlike iNTS disease in Africa, only a small fraction of Colombian bloodstream infections were caused by ST313. The majority (90.8%) of the bloodstream isolates were *S.* Typhimurium ST19.

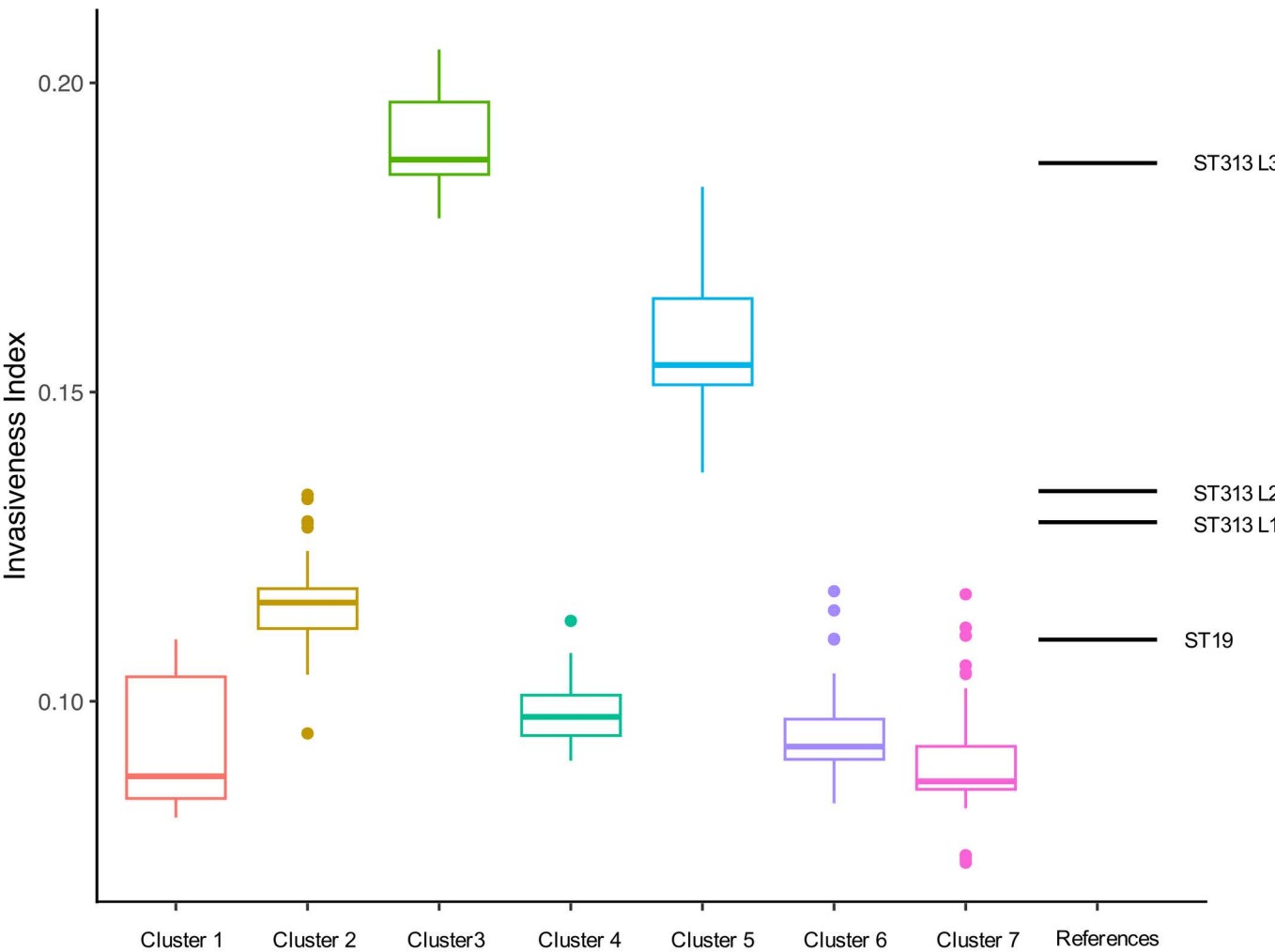

**Fig 6. The invasiveness index of 7 major Colombian HC100 clusters.** Box plot representing the distribution of invasiveness index values for the genome sequences included in this study, summarized by lineage assignment. The 7 Colombian HC100 clusters are defined in Fig 1. Number of isolates in each group: Cluster 1 (n = 10), Cluster 2 (n = 75), Cluster 3 (n = 5), Cluster 4 (n = 14), Cluster 5 (n = 7), Cluster 6 (n = 31), and Cluster 7 (n = 70). The central line in the box plot represents the median value (50th percentile), the box limits denote the 25th and 75th percentiles, the whiskers extend by 1.5 times the interquartile range and individual points represent outliers. For comparative purposes, the median invasiveness indices of African *S.* Typhimurium ST19, ST313 L1, L2 and L3 (reported in [12]) are plotted as horizontal lines on the right side of the figure. The plot was made with R package ggplot2 [52].

In Africa, ST313 was responsible for the majority of iNTS infections. For example, in Malawi, only 5.6% of the iNTS-associated *S.* Typhimurium isolates were ST19, while 94.4% were ST313 or ST313 single-locus variants [12]. However, in this study, the majority of bloodstream infection isolates in Colombia were ST19. Two global epidemic *S.* Typhimurium clusters, the DT104-associated cluster (Cluster 2) and the LT2-related cluster (Cluster 7) were associated with bloodstream infections before 2010 and after 2010, respectively. The two ST19 clusters exhibited a relatively low invasiveness index (Fig 6). In contrast, relatively few ST313 (HC100_20801, n = 5) and monophasic ST19 (HC100_1343, n = 7) isolates with a high invasiveness index were identified.

In sub-Saharan Africa, highly invasive *S.* Typhimurium ST313 lineages target immuno-compromised individuals such as infants suffering from malaria or malnutrition, or adults with HIV [8]. We hypothesised that Colombia has a significantly lower incidence of iNTS than

in SSA [9] because the Latin American country has fewer vulnerable immunocompromised people.

Data from www.unaids.org (accessed Feb 2023) was used to compare the proportion of immunocompromised adults in Colombia with SSA. The estimated rate of HIV infection among 15–49 year-old adults in Colombia was 0.5%, significantly lower than seen in SSA countries like Malawi (7.7%) or Uganda (5.2%). Here, we investigated iNTS isolates ranging from an earlier period (1997 to 2017); unfortunately, data regarding the HIV status of individual Colombian participants in this study were not available due to data confidentiality issues.

We speculate that the lower incidence of iNTS and the limited prevalence of *S.* Typhimurium ST313 in Colombia compared to in SSA reflects a combination of the pathogen and host factors: the genetic adaptation of *S.* Typhimurium ST313 to the extraintestinal lifestyle in vulnerable people and the immunocompetent of the Colombia population. A similar situation can be found in Southeast Asia and South Asia, where the incidence of iNTS and HIV-associated cases is significantly lower than in SSA [9], and the proportion of ST313 isolates is limited [70].

On the African continent, the dissemination and replacement of iNTS lineages was associated with AMR status. For instance, the replacement of the African *S.* Typhimurium ST313 lineage L1 by L2 during 2001–2004 was linked to the acquisition of the *cat* chloramphenicol resistance gene [58]. Similarly, the disparate bloodstream-associated *S.* Typhimurium clusters we identified in Colombia had a variety of antimicrobial resistance patterns. The majority of *S.* Typhimurium isolates in the DT104-associated Cluster 2 exhibited resistance to ampicillin, streptomycin, tetracycline, and sulphonamide. Certain isolates from Cluster 2 also demonstrated resistance to chloramphenicol, trimethoprim, or azithromycin (Fig 2). Subsequently, after 2010, Sub-cluster 7.3 acquired quinolone resistance encoded by *qnrB19*. The replacement of Cluster 2 by Cluster 7 in around 2010 was associated with the acquisition of quinolone resistance and a reduced incidence of ampicillin resistance. The rise of fluoroquinolone-resistant Sub-cluster 7.3 warrants close monitoring by Colombia's public health institutions.

It has been established that *Salmonella* serovars have been distributed worldwide via the food chain [57,71,72]. Consequently, we conducted an investigation to identify potential food exports responsible for the international spread of the HC50_64733 cluster from Colombia. Poultry makes up 45.5% of Colombia's animal product exports to Ecuador, according to data from MIT's Observatory of Economic Complexity (https://oec.world/en/profile/country/col). We speculate that the transmission of *Salmonella* from Colombia to Ecuador could be linked to the movement of poultry products. However, the transmission may also be driven by the movement of other food products or human travel.

With the development of industrial food production and international logistics in recent decades, the transmission of salmonellosis has become an international issue [73–75]. Genomic data from low- and middle-income countries such as Colombia suggest that the local *Salmonella* lineages are either a consequence of global transmission or represent the initiation of a transmission chain to other countries. The utilization of public databases, such as Enterobase allowed us to track *Salmonella* lineages between countries.

However, it is important to note that the genomic data present in Enterobase suffers from significant sample bias, with a skew towards genomes originating from gastroenteritis-associated *Salmonella* in high-income countries (particularly in Europe and North America). Our study helps to bridge this gap by incorporating genomic data obtained through systematic surveillance in Colombia, providing valuable insights into *Salmonella* Typhimurium within Latin America. As our research focused on human bloodstream-derived isolates rather than stool or environmental samples, its broader applicability to public health strategies in Colombia is limited. In future, international collaborations are needed to build capacity and to improve the sequencing capabilities of low- and middle-income countries [19].

## Supporting information

**S1 Table..  The metadata and accession numbers of the Colombian *Salmonella* Typhimurium used in this study.**
(XLSX)

**S2 Table..  The metadata and accession numbers of the contextual *Salmonella* Typhimurium genomes and reference plasmids.**
(XLSX)

**S3 Table..  The metadata and accession numbers of the global *Salmonella* Typhimurium genomes downloaded from Enterobase.**
(XLSX)

**S1 Fig..  The detailed Raxml tree showing monophasic Typhimurium isolates identified from 3 different methods: experimental serovar identification, the *in-silico* prediction by SISTR, and the presence or absence of the *fljAB* genes.** The coloured rings show the serotyping from different methods. Branches with bootstrap values less than 80 were marked in grey.
(PDF)

## Author contributions

**Conceptualization:** Yan Li, Caisey V. Pulford, Blanca M. Perez-Sepulveda, Jay C. D. Hinton.

**Data curation:** Yan Li, Caisey V. Pulford, Paula Díaz, Blanca M. Perez-Sepulveda, Carolina Duarte, Magdalena Wiesner, Ross Low, Angeline Montaño, Jaime Moreno.

**Formal analysis:** Yan Li.

**Funding acquisition:** Darren Heavens, Neil Hall, Jay C. D. Hinton.

**Investigation:** Yan Li, Darren Heavens, Christian Schudoma.

**Methodology:** Yan Li, Caisey V. Pulford, Alexander V. Predeus.

**Project administration:** Yan Li, Blanca M. Perez-Sepulveda.

**Resources:** Paula Díaz, Blanca M. Perez-Sepulveda, Carolina Duarte, Magdalena Wiesner, Angeline Montaño, Jaime Moreno.

**Software:** Yan Li.

**Supervision:** Alexander V. Predeus, Neil Hall, Jay C. D. Hinton.

**Validation:** Yan Li.

**Visualization:** Yan Li.

**Writing – original draft:** Yan Li, Neil Hall, Jay C. D. Hinton.

**Writing – review & editing:** Yan Li, Jaime Moreno, Jay C. D. Hinton.

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
