## [Decision Letter · Decision Letter 0]

9 Sep 2024

Dear Professor Hinton,

Thank you very much for submitting your manuscript "Potential links between human bloodstream infection by *Salmonella enterica* serovar Typhimurium and international transmission to Colombia" for consideration at PLOS Neglected Tropical Diseases. As with all papers reviewed by the journal, your manuscript was reviewed by members of the editorial board and by several independent reviewers. In light of the reviews (below this email), we would like to invite the resubmission of a significantly-revised version that takes into account the reviewers' comments. 

Please find below feedback from three reviewer's who each found your work to be of interest. They have provided detailed comments on your manuscript, which I would like you to address and incorporate into a revised version. In particular, please address comments relating to public availability of your sequencing data and ensure the data is available from the quoted SRA BioProject.

We cannot make any decision about publication until we have seen the revised manuscript and your response to the reviewers' comments. Your revised manuscript is also likely to be sent to reviewers for further evaluation.

Sincerely,

Ben Pascoe

Academic Editor

Georgios Pappas

Section Editor

Please find below feedback from three reviewer's who each found your work to be of interest. They have provided detailed comments on your manuscript, which I would like you to address and incorporate into a revised version. In particular, please address comments relating to public availability of your sequencing data and ensure the data is available from the quoted SRA BioProject.

Reviewer's Responses to Questions

**Key Review Criteria Required for Acceptance?**

**Methods**

-Are the objectives of the study clearly articulated with a clear testable hypothesis stated?

-Is the study design appropriate to address the stated objectives?

-Is the population clearly described and appropriate for the hypothesis being tested?

-Is the sample size sufficient to ensure adequate power to address the hypothesis being tested?

-Were correct statistical analysis used to support conclusions?

-Are there concerns about ethical or regulatory requirements being met?

Reviewer #1: See below for the specific points

Reviewer #2: The methods are clearly written with the study design appropriate. The authors have undertaken multiple different analyses. There were a few areas where additional clarity would be useful 

- I have no concerns about the ethics but I couldn’t find a statement. I would assume these fall under the 10,000 genome project and/or were isolates collected under public health acts / in the public domain. The authors could include a sentence to address this. 

- For the AMR genotype phenotype concordance analyses with the authors reporting phenotypes for multiple drug classes. However, there doesn’t seem to be statistical analysis e.g of the sensitivity / specificity of the AMR genotype phenotype profiles. From Fig 3 it looks to have high concordance but this was from visually look at the heatmap. 

- ML trees – these were inferred from SNPs in core gene alignments from Panaroo. Was the default flag for core used in Panaroo and what tool was used for the alignment? MAFT? Were the core gene numbers similar / different between the different trees inferred?

- Identify circularised contigs from draft assemblies. This was important for the subsequent plasmid analyses. Do you mean a single contig that circularised? Or several contigs that were fell out in the assembly graph? Additionally, I think there should be an additional cite for plsdb (https://ccb-microbe.cs.uni-saarland.de/plsdb/). I hadn’t heard of this tool before and wondered what the difference was compared to e.g. Mob-suite? 

- HeirCC was important in to this story but there was limited explanation of how the scheme in enterobase works. E.g that is cgMLST based and what the different HC levels mean (HC100 v HC50). Also how the levels relate to STs. E.g. HC100_20801 was the ST313 with sublineages in it? 

- In both methods and results, the authors used different names for the clusters (HierCC, cluster and ST). At times I had keep referring back to Fig 1 to work out what cluster was being discussed. 

- Bactdating – default parameters in the R package?

Reviewer #3: (No Response)

**Results**

-Does the analysis presented match the analysis plan?

-Are the results clearly and completely presented?

-Are the figures (Tables, Images) of sufficient quality for clarity?

Reviewer #1: the results presented in a well-standard form.

Reviewer #2: The results matched the analysis plan with the results clearly presented. There were a few things I think could be expanded upon that would strengthen the manuscript

- The 20 unknown STs – several looked to be part of HierCC_100_305. Were these SLV of the main ST of this group?

- For the AMR results, there were a few isolates that had 3GC mechanisms detected and I think were phenotypically resistant as well. Were these in the same lineages/ clusters or spread across the population? I think it is important to report on these mechanisms even if at low frequency in the dataset, as it could be the detection of emerging resistance to this important drug class in Latin America (also a priority for the WHO). Related – please cite the WHO Bacterial Priority List for 2024 instead of 2017. I don’t think colistin or carbapenem resistance was detected in the Colombian dataset. Again I think it is useful to note the absence of AMR mechanisms to these drug classes as well. 

- In reporting the point mutations in QRDR regions associated with quinolone resistance, please report the specific mutations (page 15 and supp table quinolone tab), especially as combinations of these point mutations are associated with different phenotypes (e.g. 1 point vs 3). Were any point mutations in parC codon 80 detected?

- Plasmid presence – Fig 3. The heatmap shows plasmid cov including for the pCOT7 which looks to have ~0.4 coverage. I couldn’t see the details of this plasmid in Supp table 2 reference plasmid tab – I was looking as I wanted to know the plasmid length. Would the authors consider it important to have a threshold for the plasmids in this analysis? E.g > 0.8 coverage of plasmid to have confidence in that plasmid (or something close) in the population. In Fig 3 – it was difficult to distinguish between 0.8 and 1 coverage. Suggest changing the colours to make this easier to interpret. 

- There was some text in the results that I think belong in the methods. E.g. first para of page 18 “To identify the non-pSLT..”

- Numbers for the different clusters. In the text HC100_20801 has 87 isolates but the Supp Table 3 had 95 isolates- why was there a difference?

- Some text in results would be better in the discussion / conclusions. E.g last para of page 23 discussing the lack of animal models. 

- For the invasive index analyses – it was hard to determine the significance of the index in the population. There were differences but from Fig 6, all values were <0.20 but in the previous para on page 24 the authors state that a high index is a signal these bacteria are adapting to extraintestinal lifestyle. Do the authors know how e.g. Typhi performs with this index? Is 0.1 high?

Reviewer #3: (No Response)

**Conclusions**

-Are the conclusions supported by the data presented?

-Are the limitations of analysis clearly described?

-Do the authors discuss how these data can be helpful to advance our understanding of the topic under study?

-Is public health relevance addressed?

Reviewer #1: see the specific points.

Reviewer #2: The conclusions are largely supported by the results. Additional clarification of methods and results will help. The public health relevance was clearly addressed. 

- Analysis on HIV infection and trying to explore the host factors was reported in the discussion / conclusion (page27). Should this be included in methods and results? Or was the point that it was difficult to explore host factors.

- The food exports section is interesting but it is possible it could be the movement of people that then spread the bacteria to livestock. Suggest rephrasing this text a bit. 

- A paragraph on the strengths and limitations of this study would be valuable. For example, a key strength is the dataset from a region where to date there has been a paucity of data (e.g. in the recent Carey 2023 eLIFE on Typhi there was limited isolates from Latin America). A limitation would be the sampling biases in the public data and some consideration as to how this may impact findings based on sampling approaches).

Reviewer #3: (No Response)

**Editorial and Data Presentation Modifications?**

Reviewer #1: all the new data should be free in the public domain.

Reviewer #2: - The introduction had only 8 references and some of these were dated. E.g. an important genomic study of monophasic salmonella was the Petrovska 2016 EID and this wasn’t cited until the discussion. There was a bias in references to those from the UK. Additional references eg from Europe, North America for implementation of routine WGS and use of genomics to explore typhimurium and monophasic in settings in Africa and Asia (e.g Van Puyvelde 2023 Nat Comms for iNTS in sub Saharan Africa and The 2023 Comms Bio for MDR plasmids in South East Asia) would provide greater context for this study especially given the focus on global spread of the pathogen

- Define MDR in the introduction – it is explained in the results

- Explain what iNTS is and what is meant by ‘systematic iNTS disease’. This is really important to the underlying rationale for this study 

- The authors refer to DT104 throughout the manuscript. I think this relates to phage typing that used to be done for Salmonella? Some explanation of changes to surveillance efforts for NTS and how genomics has changed it etc would be really useful in the introduction. This would help the broad readership of PLoS NTD know what DT104 means and why it is important (or has this nomenclature been replaced?)

Reviewer #3: (No Response)

**Summary and General Comments**

Reviewer #1: The study provides valuable insights into the genomic characteristics of Salmonella Typhimurium isolates responsible for bloodstream infections in Colombia, particularly emphasizing the identification of ST19 as the predominant sequence type. The investigation into the potential international transmission and the comparison with global isolates adds significance to the findings. However, the novelty of the study could be better emphasized by providing a deeper discussion on how these findings could influence public health strategies in Colombia and similar regions.

Major points

Phylogenetic Analysis: The phylogenetic analysis appears robust, linking Colombian isolates to global clusters. However, the manuscript would benefit from a more detailed explanation of the methods used for phylogenetic tree construction and statistical support (e.g., bootstrap values). Additionally, while the study identifies multiple independent transmission events, it would be helpful to discuss potential sources or routes of these transmissions in more detail.

Plasmid and Antimicrobial Resistance (AMR) Analysis: The study highlights the presence of MDR profiles and plasmid-associated resistance. While this is well documented, the manuscript could provide more information on the functional impact of these plasmids. For instance, how do these plasmids contribute to the fitness of S. Typhimurium in the host environment? Are there any potential targets for novel therapeutics identified in these plasmids?

ST313 in Colombia: The identification of ST313 in Colombia, a sequence type predominantly associated with sub-Saharan Africa, is a crucial finding. The study discusses its limited prevalence in Colombia, which is attributed to the immune-competence of the population. However, this assertion could be further supported by epidemiological data or references to similar studies in other regions with immune-competent populations. Moreover, it would be valuable to discuss the potential public health implications if ST313 were to become more prevalent.

Data Interpretation and Public Health Implications: The manuscript could expand on the public health implications of these findings. How can this genomic information be used to enhance current surveillance systems in Colombia? Are there specific recommendations for public health policy that could arise from this study, such as the monitoring of particular MDR strains or the implementation of targeted interventions?

Comparative Analysis with Global Isolates: The manuscript does a good job of placing the Colombian isolates in a global context. However, the discussion could be strengthened by providing more comparative insights between the Colombian isolates and those from other regions, particularly in terms of genomic variations and their potential functional consequences.

Minor Points

The methods section, while comprehensive, could benefit from a clearer description of the bioinformatic tools and parameters used. For example, the criteria for defining clusters and the thresholds for AMR gene identification should be explicitly stated.

Figures and Tables:

The figures and tables are informative but could be more effectively used to highlight key findings. For instance, a figure summarizing the phylogenetic relationships alongside plasmid and AMR profiles would provide a visual overview of the study's main points. Additionally, more detailed legends explaining the figures would help readers interpret the data more easily.

While the manuscript is well-referenced, it could incorporate more recent studies on Salmonella genomics, particularly those focusing on Latin American countries. This would help place the findings in a broader context and demonstrate how they advance the current understanding of Salmonella epidemiology.

The manuscript is generally well-written, but some sentences could be revised for clarity. For instance, the explanation of the significance of ST19 versus ST313 in the Colombian context could be more concise and directly linked to the study’s conclusions.

These comments should provide a balanced perspective on the manuscript, acknowledging its strengths while offering constructive suggestions for improvement. If further detailed analysis or additional specific critiques are needed, please let me know.

Reviewer #2: Overall this was an interesting and well-presented study on a topical area. The new sequence data in this study will be of use to researchers global given the limited data on Latin America to date. 

Data availability

- The authors have made a very good effort at making the data available in this study. In the data availability statement it says the accessions are available in S1 table. I could see the barcode and strain name in the Main tab but not the accession. Please add the SRR/ERR for each isolate. 

- I looked for PRJNA1095721 but couldn’t find this BioProject. Please make public. It was unclear if the complete genome sequences of the novel plasmids were from the short read data reported in this study. Or if ONT data was used. Please make this clear and if ONT data was used – additional methods need to be included.

Reviewer #3: In this study, Li et al report and analyze genomic characteristics of a library of S. Typhimurium strains isolated from individual with bloodstream infections in Colombia. Whole genome sequencing for 270 strains is performed, and the phylogeny of these strains (including various reference strains) was performed. Resistance to various antimicrobials is predicted from the genome assemblies, and confirmed phenotypically. Plasmid content and distribution is analyzed as well. The most interesting part of the study is the attempt at retracing global transmission routes of strains isolated from Colombia. Cluster 2 strains (MDR DT104-related) were apparently introduced at least three different times from different sources. Quinolone/chloramphenicol resistant strains originated in Colombia and were introduced to other countries. Cluster 3 (invasive NTS-associated ST313, originally identified in Africa) was introduced to Colombia through at least two distinct events. Invasiveness of strains was predicted based on genomic content. 

Overall, this study addresses an important need. Most studies on the genomics of Salmonella (invasive and gastroenteritis-associated strains) are done on isolates from Europe and the US, due to sampling bias. In this study, the authors focused on Colombian isolates, which is a strength of this study. The most interesting finding is the (apparently) frequent exchange of S. Typhimurium isolates with other countries, as shown in Fig. 4 and 5. Main conclusions are justified, the reading is clear, and the data are presented well. I have no major concerns. Some minor comments for the authors’ consideration are below.

Minor comments, edits:

There are a lot of abbreviations used in the main sections. Some of them are unavoidable, such as strain designations, etc. To improve clarity, I would suggest to keep all other abbreviations to a minimum. For example, SSA (sub-saharan Africa), iNTS, AMR, MDR.

Pge 27: “Here. We investigated”

PLOS authors have the option to publish the peer review history of their article (what does this mean? ). If published, this will include your full peer review and any attached files.

**Do you want your identity to be public for this peer review?** For information about this choice, including consent withdrawal, please see our Privacy Policy .

Reviewer #1: Yes: min yue

Reviewer #2: No

Reviewer #3: No
---

## [Editor Report · Decision Letter 1]

19 Dec 2024

Dear Professor Hinton,

We are pleased to inform you that your manuscript 'Potential links between human bloodstream infection by *Salmonella enterica* serovar Typhimurium and international transmission to Colombia' has been provisionally accepted for publication in PLOS Neglected Tropical Diseases.

Best regards,

Ben Pascoe

Academic Editor

Georgios Pappas

Section Editor

Shaden Kamhawi

co-Editor-in-Chief

Paul Brindley

co-Editor-in-Chief

Thank you for responding thoroughly to the concerns raised by the reviewers and incorporating their feedback into a revised version of the manuscript.

---

## [Editor Report · Acceptance letter]

Dear Professor Hinton,

We are delighted to inform you that your manuscript, "Potential links between human bloodstream infection by *Salmonella enterica* serovar Typhimurium and international transmission to Colombia," has been formally accepted for publication in PLOS Neglected Tropical Diseases.

Best regards,

Shaden Kamhawi

co-Editor-in-Chief

Paul Brindley

co-Editor-in-Chief
